# Production of leishmanin skin test antigen from *Leishmania donovani* for future reintroduction in the field

Ranadhir Dey [1,8], Jalal Alshaweesh[2,3,8], Kamaleshwar P. Singh[4,8], Patrick Lypaczewski [5], Subir Karmakar[4], Laura Klenow[1], Kayla Paulini [5], Swarnendu Kaviraj[4], Shaden Kamhawi [6], Jesus G. Valenzuela [6], Sanjay Singh[4] ✉, Shinjiro Hamano [2,3] ✉, Abhay R. Satoskar [7] ✉, Sreenivas Gannavaram [1] ✉, Hira L. Nakhasi [1] ✉ & Greg Matlashewski [5] ✉

The leishmanin skin test was used for almost a century to detect exposure and immunity to *Leishmania*, the causative agent of leishmaniasis, a major neglected tropical disease. Due to a lack of antigen used for the intradermal injection, the leishmanin skin test is no longer available. As leishmaniasis control programs are advancing and new vaccines are entering clinical trials, it is essential to re-introduce the leishmanin skin test. Here we establish a *Leishmania donovani* strain and describe the production, under Good Laboratory Practice conditions, of leishmanin soluble antigen used to induce the leishmanin skin test in animal models of infection and vaccination. Using a mouse model of cutaneous leishmaniasis and a hamster model of visceral leishmaniasis, soluble antigen induces a leishmanin skin test response following infection and vaccination with live attenuated *Leishmania major* (*LmCen*$^{-/-}$). Both the CD4$^+$ and CD8$^+$ T-cells are necessary for the leishmanin skin test response. This study demonstrates the feasibility of large-scale production of leishmanin antigen addressing a major bottleneck for performing the leishmanin skin test in future surveillance and vaccine clinical trials.

Leishmaniasis is a neglected tropical disease caused by protozoan parasites of the genus *Leishmania* that are transmitted by phlebotomine sand flies[1]. The majority of *Leishmania* species are zoonotic and cause cutaneous leishmaniasis (CL) that manifests as a localized skin lesion at the site of the vector sand fly bite that typically but not always heals spontaneously after several weeks or months. Visceral leishmaniasis (VL) caused by *L. donovani* is anthroponotic and involves the dissemination of the parasite to the reticulo-endothelial system including the liver, spleen and bone marrow resulting in internal hemorrhaging, severe anemia, co-infections due to immunosuppression and is fatal if not treated. Cases of CL and VL are escalating in different parts of the world due to the increasing migration of populations into and from endemic areas and changes in ecosystems[1]. There is no vaccine for human leishmaniasis, and all currently used drugs were initially developed for other indications and often have poor efficacies depending on the species and disease pathology[2].

[1]Division of Emerging and Transfusion Transmitted Diseases, CBER, FDA, Silver Spring, MD, USA. [2]Department of Parasitology, Institute of Tropical Medicine (NEKKEN), Nagasaki University, Nagasaki, Japan. [3]The Joint Usage/Research Center on Tropical Disease, Institute of Tropical Medicine (NEKKEN), Nagasaki University, Nagasaki, Japan. [4]Gennova Biopharmaceuticals, Hinjawadi Phase II, Pune, Maharashtra, India. [5]Department of Microbiology and Immunology, McGill University, Montreal, QC, Canada. [6]Vector Molecular Biology Section, Laboratory of Malaria and Vector Research, National Institute of Allergy and Infectious Diseases, NIH, Rockville, Maryland 20852, USA. [7]Department of Pathology and Microbiology, Ohio State University, Columbus, OH, USA. [8]These authors contributed equally: Ranadhir Dey, Jalal Alshaweesh, Kamaleshwar P. Singh ✉e-mail: Sanjay.Singh@gennova.co.in; shinjiro@nagasaki-u.ac.jp; Abhay.Satoskar@osumc.edu; Sreenivas.Gannavaram@fda.hhs.gov; hira.nakhasi@fda.hhs.gov; greg.matlashewski@mcgill.ca

One of the significant developments in the field has been the VL elimination program initiated in 2005 in India, Nepal and Bangladesh aiming to reduce the disease incidence to less than one case per 10,000 population at the sub-district level. The elimination strategy is mainly based on improved case management, integrated vector control and effective disease surveillance[3]. The VL incidence in these countries has dropped significantly coinciding with the elimination initiative, although it is difficult to assess whether the natural periodicity of the disease has also contributed to the decline in cases[3,4]. Most infections do not progress to disease remaining asymptomatic, and there is a long incubation period before symptomatic VL does develop. Monitoring asymptomatic infection rates in addition to VL cases would therefore provide more accurate information on transmission and strengthen ongoing country-wide control programs.

Leishmaniases are potentially vaccine-preventable diseases since cure following treatment results in acquired immunological protection against re-infection[5]. Currently, there are vaccines that are soon to enter clinical trials such as the live attenuated *LmCen*[−/−] vaccine[6] or the ChAd63-KH Adenovirus-based vaccine that is currently undergoing clinical trials[7]. Major considerations for future human VL vaccination studies include the clustering and geographically changing location of cases, the reduction in cases due to the elimination program in South Asia and nomadic susceptible populations in East Africa collectively making it difficult to conduct clinical trials based on case numbers. The availability of a controlled human infection model will help address this issue[8], but it is nevertheless necessary to carry out trials in endemic areas involving larger populations. It will therefore be necessary to identify a surrogate marker of protective immunity for future vaccine trials.

The leishmanin skin test (LST), otherwise known as the Montenegro skin test[9] could be used to both improve transmission surveillance and as a biomarker for vaccine efficacy. The LST is performed via intradermal injection of *Leishmania* antigens (leishmanin) into the forearm to visualize the adaptive cellular immune response in individuals who have been previously infected with *Leishmania*[10]. The test is analogous to the Mantoux tuberculin skin test (TST) which is widely used as a diagnostic test for tuberculosis. Both the LST and TST are based on the principle that intradermal injection of antigens into an immune individual causes a T-cell mediated delayed-type hypersensitivity (DTH) response. The DTH response is "delayed" because the maximal influx of T-helper cells and other inflammatory cells occurs 24–72 h after exposure to antigens[11] where an induration of at least 5 mm in diameter is considered a positive test demonstrating previous exposure to *Leishmania*[12]. Since the protective immune response to *Leishmania* is primarily mediated by T-helper 1 cells[13], a positive LST also indicates immunity against infection. Consequently, epidemiologic studies have shown that a positive LST is associated with long-lasting protective immunity against VL[14–16] and is mediated by both CD4[+] and CD8[+] T-cells[17,18]. The LST could therefore be used to monitor transmission or as a surrogate marker of immunity in vaccine clinical trials. The LST is better than serology for determining exposure to *Leishmania* because cell-mediated immunity lasts for decades compared to the antibody response that lasts for months and further, the LST is more indicative of immunological protection against reinfection[10]. Despite efforts over the past few decades, the leishmanin antigen used for the LST is currently not produced under good manufacturing practice (GMP) conditions anywhere in the world and therefore the LST is no longer available[10].

In this work, we extract soluble leishmanin antigen from a highly characterized virulent strain of *L. donovani*. The soluble leishmanin antigen is used to validate the LST response in infected and vaccinated mice and hamsters. This study represents the foundation for the production of Good Manufacturing Practice (GMP)-grade leishmanin antigen to re-establish the LST for human surveillance and vaccination programs in endemic countries.

## Results

### Analysis of an LST response against soluble leishmanin antigen in immune C57BL/6 mice

Similar to humans, C57BL/6 mice infected with *L. major* parasites clear the infection and are subsequently immune to reinfection[19]. Thus, this represents a good animal model to investigate the LST response against a leishmanin soluble antigen derived from *L. donovani*. Further, it is possible to determine if vaccination with the live attenuated *L. major* parasites (*LmCen*[−/−]) can stimulate a similar LST response. As outlined in Fig. 1A, mice were inoculated intradermally with 1000 wild-type *L. major* FV9 (leishmanized) and infected lesions were allowed to heal over 12 weeks. Alternatively, mice were inoculated $1 \times 10^6$ *LmCen*[−/−] live attenuated *L. major* (vaccinated) that provided immunity to re-infection without lesion development[6]. Infected and naïve mice were then injected intra-dermally in the contralateral ears with 5ug of soluble leishmanin antigen prepared from *L. donovani* Ld-Ind, and the ears were analyzed at 24 and 48h post-injection. Soluble antigen was prepared by freeze-thaw cycles as described in Methods, and the SDS-PAGE shows that the soluble protein ranged from 120 to 25 kD (Fig. 1B).

As shown in representative animals (Fig. 1C), naïve mice that received only excipient media did not show any LST response at 24 and 48h post-inoculation whereas mice infected with either wild-type *L. major* or attenuated *L. major LmCen*[−/−] showed measurable and significant LST responses represented by the induration and erythema of the ear dermis (Fig. 1C, D). Hematoxylin and eosin (H&E) staining of the ear dermis inoculation site revealed cellular infiltration in the tissue (Fig. 1E, F). Immuno-histological staining revealed that the LST response was associated with an influx of CD4[+] and CD8[+] T-cells in the mice infected with wild-type *L. major* or vaccinated with attenuated *LmCen*[−/−], while the naïve mice showed only background levels (Fig. 1G, H).

It is necessary for a standardized LST to be reproducible in different settings in the field. The LST response was therefore performed in a second independent laboratory at Nagasaki University, Japan to test the robustness and reproducibility of the LST under the same protocol (Supplementary Fig. 1A). Immune C57BL/6 mice were injected with soluble antigen prepared independently from *L. donovani* promastigotes and the LST response was measured 24 and 48h post-inoculation. As shown in Supplementary Fig. 1B–D, the LST response was detectable in an independent lab with their antigen preparation and inoculation of C57BL/6 mice under the same protocol confirming reproducibility in laboratories in the USA and Japan.

### Determining the role of CD4[+] and CD8[+] T-cells in the LST response

Figure 1 shows the infiltration of both CD4[+] and CD8[+] T-cells into the dermal site of the LST response. We therefore examined the contribution of each CD4[+] and CD8[+] T-cell subset in the LST response by depleting them. The first group of mice was infected with a low dose, 1000 parasites/mouse of virulent *L. major* (FV9) parasites. The inoculated mice cleared the infection after 12 weeks (leishmanized, Fig. 2A). The second group of mice was immunized with $1 \times 10^6$ *LmCen*[−/−] attenuated parasites for 12 weeks (vaccinated, Fig. 2A). Depleting CD4[+] or CD8[+] T-cells was accomplished with a single injection of either anti-CD4 mAb or anti-CD8 mAb as detailed in the Methods. The cell depletion was confirmed in sampled mice using flow cytometry and Rag2[−/−] mice were used as a control for the depletion of CD4[+] and CD8[+] T-cells (Fig. 2B, Supplementary Fig. 2). At 36h post-depletion, soluble antigen was injected into the skin dermis of the contra-lateral ear and the LST response evaluated at 24 and 48h post-injection (Fig. 2A). Infected and vaccinated mice that did not receive depleting antibodies (Saline) showed a measurable and significant LST response as expected at 24 and 48h (Fig. 2C–F). However, either CD4[+] or CD8[+] T-cell depletion in infected and vaccinated mice had significantly decreased LST reactions with less erythema and thickness (Fig. 2C–F). H&E

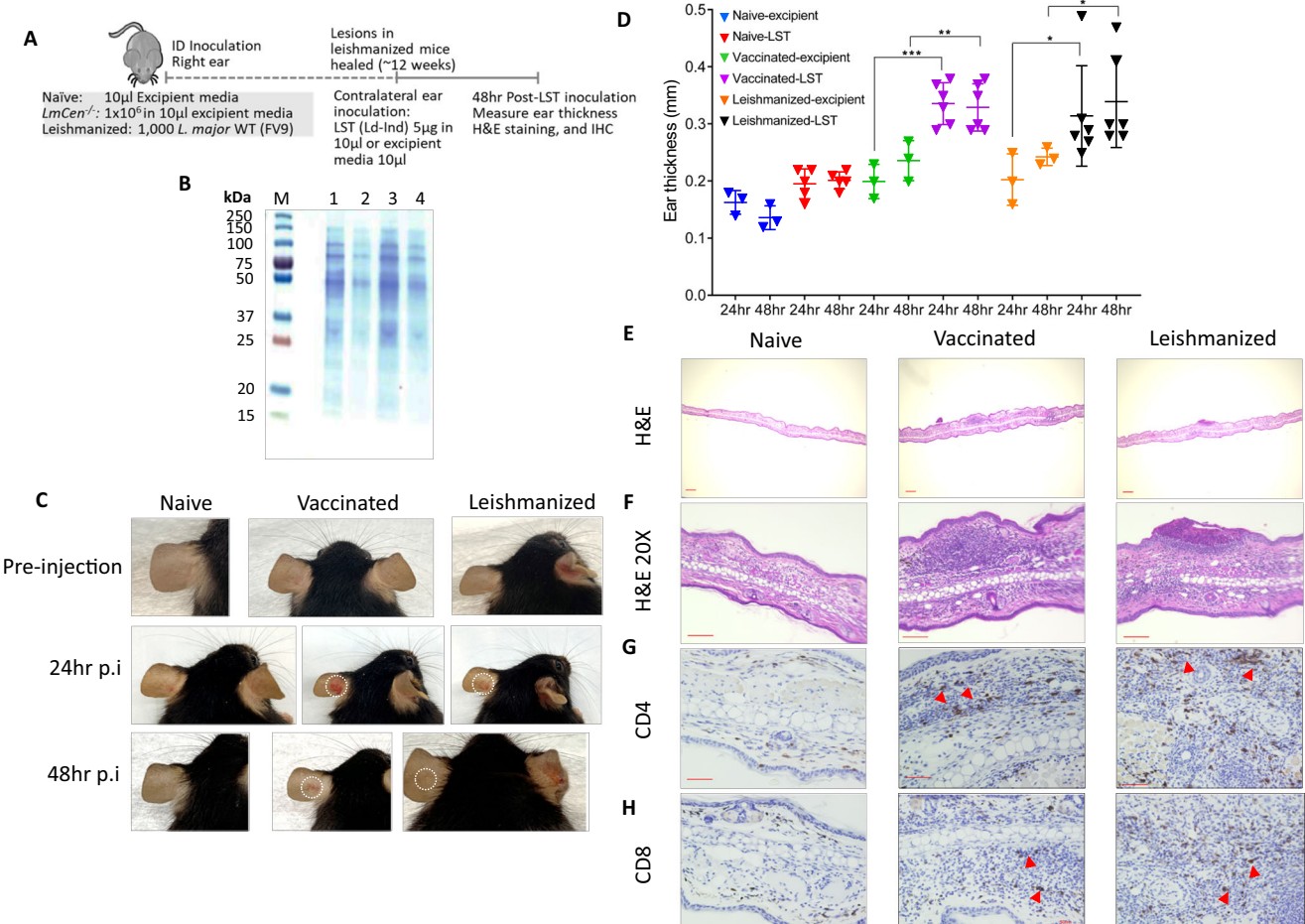

**Fig. 1 | Testing research grade leishmanin antigen as a surrogate of immunogenicity in a mouse model of cutaneous leishmaniasis. A** Schematic diagram showing the schedule of LST in mice either leishmanized with wildtype *L. major* FV9 or immunized with attenuated *L. major LmCen*[−/−]. ID, intradermal. IHC, immunohistochemistry. H&E, Hematoxylin and eosin. WT, wildtype. LST, Leishmanin Skin Test. p.i., post-infection **B** SDS-gel showing antigens prepared from a virulent stock of *L. donovani* Ld-Ind as detailed in methods; protein ladder (M), lane-1 crude Ld-Ind antigen 25 µg, lane-2: crude Ld-Ind antigen 12.5 µg, lane-3: Ld-Ind soluble antigen 25 µg, lane-4: Ld-Ind soluble antigen 12.5 µg. Crude antigen represents unclarified parasite lysates. **C** Naïve, *LmCen*[−/−] immunized (vaccinated) or leishmanized mice were inoculated intradermally with soluble antigens in the contralateral ear and the LST response was monitored at 24 and 48 h post-inoculation. Erythematous skin tissue following LST is highlighted in dotted circles. **D** The ear thickness was measured in naïve, vaccinated or leishmanized mice that were inoculated with either excipient or LST antigens. *n* = 6 per group. Data are presented as mean

values ± SEM. Unpaired two tailed Student's *t* test was used to calculate statistical significance between 24 h vaccinated-excipient and 24 h vaccinated-LST groups (*p* = 0.0009), 48 h vaccinated-excipient and 48 h vaccinated-LST groups (*p* = 0.0128), 24 h leishmanized-excipient and 24 h leishmanized-LST groups (*p* = 0.0432), 48 h leishmanized-excipient and 48 h leishmanized-LST groups (*p* = 0.045), *$p \le 0.05$; **$p \le 0.01$; ***$p \le 0.001$. Results are shown as mean ± SD, (*n* = 3 for naïve-excipient, vaccinated- excipient and leishmanized-excipient groups; *n* = 5 for naïve-LST; *n* = 6 for vaccinated-LST, and leishmanized-LST groups. Data are the representation of two biologically independent experiments. Source data are provided as a Source Data file. **E** H&E-stained ear tissue sections at 1× (Scale bar 200 µm) and **F**) at ×20 magnification showing the cellular infiltration at the site of inoculation in vaccinated and leishmanized mice. Scale bar 100 µm. **G** Immunohistochemical staining of the ear sections showing the infiltration of CD4[+] and (**H**) CD8[+] T cells (red triangles) following LST at 48h post-inoculation. Scale bar 50 µm.

---

staining of the ear dermis injection site showed reduced induration and fewer cellular infiltrates after CD4[+] or CD8[+] T-cell depletion than non-depleted mice (Fig. 2G). Taken together, these observations reveal that both the CD4[+] and CD8[+] T cell populations were involved in generating the LST response to soluble antigens in the infected and vaccinated mice.

### Isolation of a virulent *L. donovani* strain through sandfly mediated infection of hamsters

Having demonstrated that the LST response could be validated in the CL-mouse animal model, it was necessary to establish, characterize and standardize a virulent *L. donovani* strain for preparation of soluble antigen under GLP conditions. To generate virulent *L. donovani* Ld-Ind parasites, a hamster model infected by sand fly was used because it recapitulates the features of pathogenesis characteristic of human VL

(Fig. 3A). *Lu. Longipalpis* sand flies were infected with cell suspension containing *L. donovani* Ld-Ind amastigotes recovered from the spleen of a moribund hamster using a rabbit blood/chicken membrane model described previously[20] (Supplementary Fig. 3). The full developmental lifecycle of *L. donovani* parasites including attachment to the midgut and differentiation into the infectious metacyclic stage was monitored by counting the parasites at various time points following sandfly infection (Fig. 3B). By day 13 of sand fly infection, about 80% of the parasites isolated from the midguts differentiated into metacyclic stages (Fig. 3B). The fitness of *L. donovani* parasites as indicated by the metacyclogenesis in the sand flies and infectiousness to hamsters was performed several times over several years to ensure maintenance of virulence characteristics (Supplementary Fig. 4A–D). The hamsters were exposed to the infected sand flies on day 15 and the development of clinical signs of VL was monitored for pathogenesis. Spleen

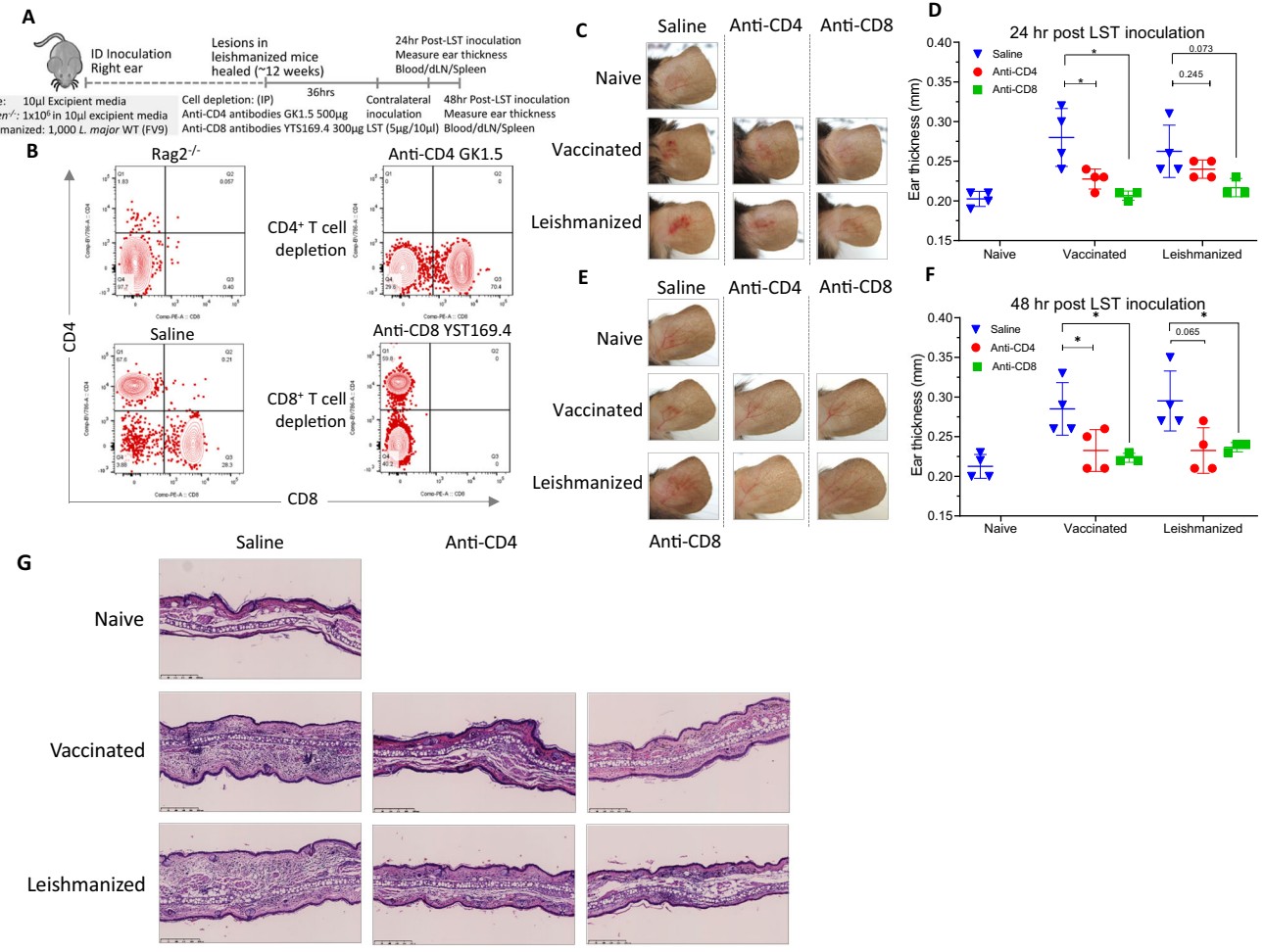

**Fig. 2 | Immune mediators of the LST response.** CD4[+] or CD8[+] T cells were depleted in the wildtype *L. major* FV9 leishmanized or the *LmCen*[−/−] vaccinated C57BL/6 mice by intraperitoneal injection of 500 μg of anti-CD4 mAb (clone GK1.5) or 300 μg of anti-CD8 mAb (clone YTS169.4), respectively. The control group (no depletion) was injected with 100 μL saline. At 36 h post-depletion, mice were injected contralaterally with 5 μg of soluble antigens, and DTH (Delayed type hypersensitivity) response was evaluated at 24 and 48h post-inoculation. **A** Illustration of the experimental design. ID, intradermal. dLN, draining lymph node. WT, wildtype. LST, Leishmanin Skin Test. **B** Representative flow cytometric dot plots of the CD3[+] splenocytes showing the depletion efficacy. **C** Photographs of mice ears taken at 24 h post-injection of LST antigens. **D** The corresponding ear thickness. Results are shown as mean ± SD, n = 3 (for Vaccinated-αCD8-LST, Leishmanized-αCD8-LST) and n = 4 (for all other conditions) biologically independent

animals. Unpaired two tailed Student's *t* test was used to calculate statistical significance between 24 h saline injected and α-CD4 treated vaccinated groups (*p* = 0.0347), 24h saline injected and α-CD8 treated vaccinated groups (*p* = 0.02). **p ≤ 0.05. **E** Photographs of mice ears taken at 48 h post-injection of LST antigens. **F** The corresponding ear thickness. Results are shown as mean ± SD, n = 3 (for Vaccinated-αCD8-LST, Leishmanized-αCD8-LST) and n = 4 (for all other conditions) biologically independent animals. Unpaired two tailed Student's *t* test was used to calculate statistical significance. 48 h saline injected and α-CD4 treated vaccinated groups (*p* = 0.0478), 48 h saline injected and α-CD8 treated vaccinated groups (*p* = 0.0265), 48h saline injected and α-CD8 treated leishmanized groups (*p* = 0.0492). **p ≤ 0.05. **G** Representative ears stained with H&E (×20) at 48h post-injection of LST antigens. Scale bars 50 μm. Source data are provided as a Source Data file.

harvested from a moribund hamster (Fig. 3C) was used for isolating and cloning *L. donovani* Ld-Ind parasites from a splenocyte suspension. Clones of *L. donovani* Ld-Ind promastigotes selected by plating on Nobel-agar plates were monitored for their growth and 3 clones with similar growth characteristics were selected for further characterization (Fig. 3D). These established *L. donovani* Ld-Ind clones have therefore cycled through the infective metacyclic stage in sand flies, infected hamsters by sand fly transmission, visceralized to the liver and spleen ultimately caused VL. These clones are therefore representative of a wild-type virulent *L. donovani* strain suitable for isolation of soluble leishmanin antigen to be used in the LST assay.

## Whole genome sequencing of virulent *L. donovani* Ld-Ind clones

Whole genome sequencing was performed to establish the genetic identity, generate a baseline sequence and confirm the geographic origin of the *L. donovani* Ld-Ind strain. Sequences for 3 clones were generated using Illumina NovaSeq deep sequencing and deposited at

GenBank (PRJNA893015). A *L. donovani* phylogeny tree was generated as described[21] using all available whole-genome *L. donovani* sequences from around the world. The *L. donovani* LdCL strain was used as a reference since this sequence is complete with no gaps[22]. This analysis shows a geographical clustering of genome sequences and reveals that *L. donovani* Ld-Ind clusters with strains derived from the Indian Subcontinent (Fig. 4A). Sequence alignment showed there were no homozygous single nucleotide polymorphisms differences between the three *L. donovani* Ld-Ind clones. No differences were identified in copy number variation at the gene or chromosome level among the 3 genomes (Fig. 4B). These observations unequivocally confirm that the three *L. donovani* Ld-Ind clones are genetically identical and derived from the Indian subcontinent.

## Production of GLP-grade soluble leishmanin antigen

It is necessary to establish production under Good Laboratory Practice (GLP) conditions for pre-clinical studies prior to production

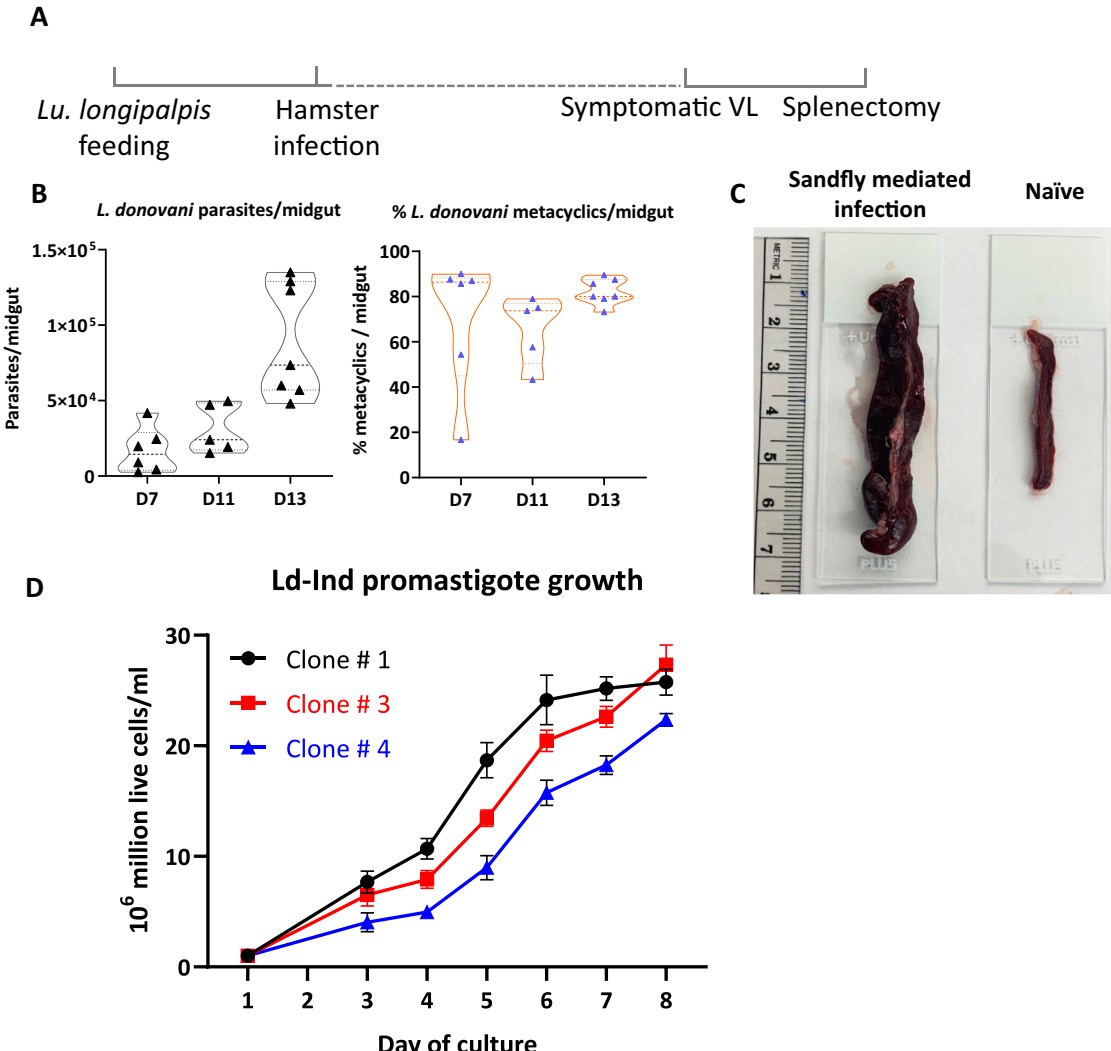

**Fig. 3 | *L. donovani* Ld-Ind Sand fly infection and recovery cloning from infected hamster spleens. A** Schematic diagram showing the schedule of membrane feeding of the *Lu. longipalpis* sandflies with spleen homogenates containing virulent amastigotes of *L. donovani* followed by infection of hamsters. VL, visceral leishmaniasis. **B** Following membrane feeding of sand flies, differentiation of amastigotes into virulent metacyclic stages was monitored by dissection of sandfly midguts at the indicated time points and microscopic counting of *L. donovani* promastigotes. The percentage of metacyclics per midgut at various growth stages of parasites is shown. **C** Infected hamsters are monitored for the development of clinicopathological symptoms of VL. Representative images of spleens from isolated from moribund hamsters and naïve uninfected control are shown. **D** Live *L. donovani* promastigotes were grown from the spleen homogenates in axenic culture and clones of parasites were isolated by plating on Nobel Agar media. Growth characteristics of three representative clones of *L. donovani* parasites are shown. Data are presented as mean values ± SEM. Source data are provided as a Source Data file.

under Good Manufacturing Practice (GMP) for human studies. We therefore established the conditions for the production of soluble leishmanin antigen under GLP compliant conditions including personnel training, qualified equipment, controlled testing facilities, record keeping and documentation. GLP grade soluble leishmanin antigen was prepared from four different lots according to the schematic diagram represented in Fig. 5A (detailed in Methods section, supplementary Fig. 5). Supplementary Fig. 5 shows the reproducibility of antigen production from 3 different lots. Promastigotes were grown in one liter of media in a bioreactor to a density of $56 \times 10^6$/ml until the cultures reached a stationary phase at 96h. The viability remained at 98% over the 96h (Fig. 5B, C,) and protein concentrations in the range of 1.32 to 2.68 mg/ml were obtained in the soluble antigen prepared from the parasites grown in the bioreactor with apparent molecular weights between 10-250 kD shown on SDS-PAGE in Fig. 5D and supplementary Fig 5B. The concentration of the soluble leishmanin antigen per one liter of culture was ~1.8 mg/ml (Fig. 5D) and which was diluted to 50 µg/

100 µl in sterile PBS containing Tween 80 (0.0005% w/v) and phenol (0.28% w/v) and used in potency tests.

### Testing of GLP-grade soluble antigen in immune hamsters

It has been demonstrated that hamsters are rendered immune against *L. donovani* by vaccination with live attenuated *L. major* LmCen[-/-23]. This represents an excellent model to establish whether the soluble antigen made under GLP conditions can generate an LST response in hamsters with immunity against *L. donovani*. Soluble antigen was therefore prepared under GLP grade conditions from the *L. donovani* Ld-Ind research cell bank (RCB) cultured at one liter in a bioreactor (Fig. 5). Hamsters infected with either live attenuated *L. major* LmCen[-/-] or wild-type *L. major* FV9 were injected intradermally with 5ug soluble antigen in the contra-lateral ear 17 weeks after infection and the LST response was measured 24 and 48h post-injection (Fig. 6A). Naïve hamsters that received only excipient media did not show any LST response, whereas both *LmCen[-/-]* vaccinated and *L. major* FV9 infected hamsters showed measurable LST responses represented by the

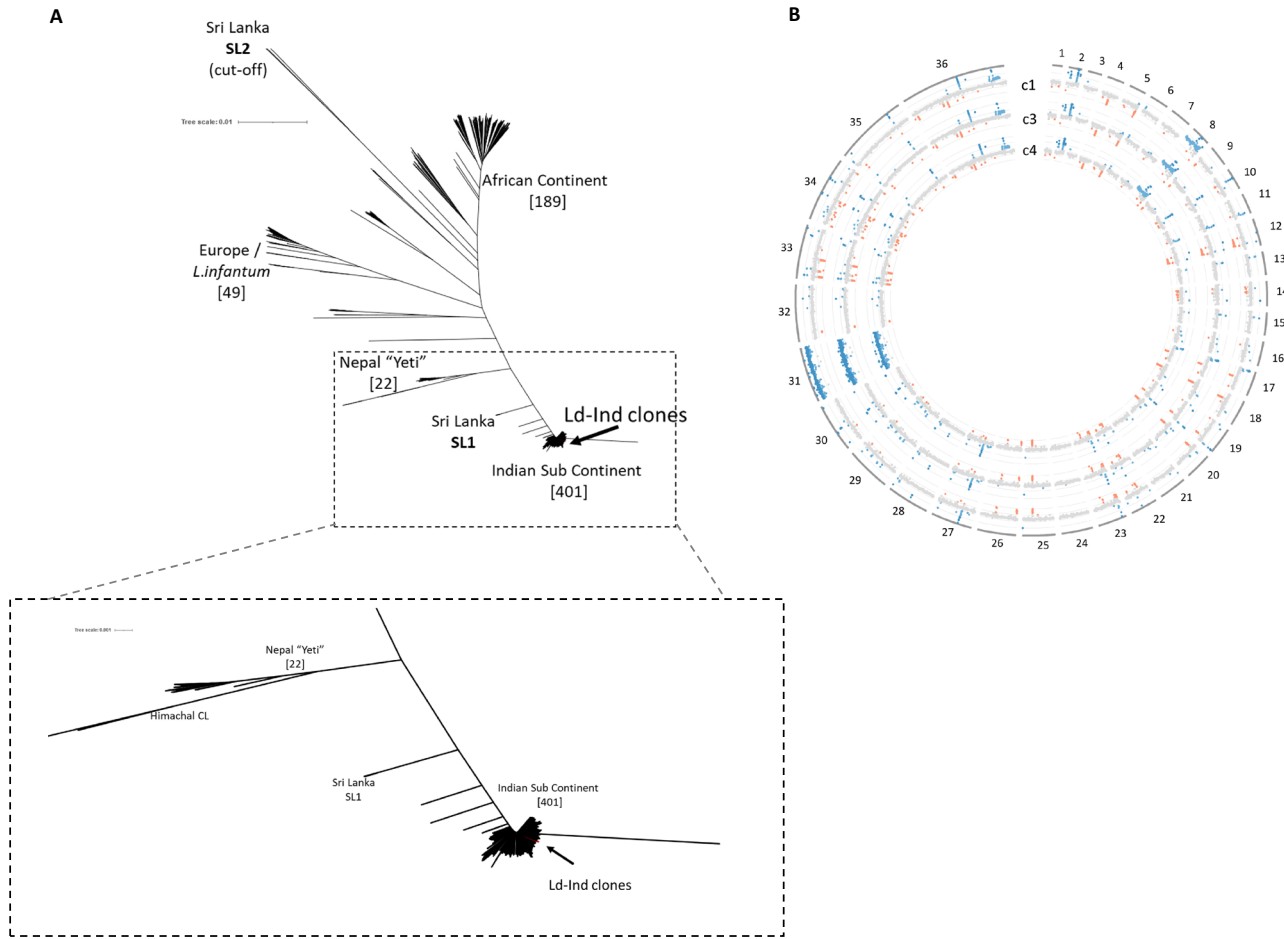

**Fig. 4 | *L. donovani* Ld-Ind- whole genome sequencing and establishment of genetic identity. A** Phylogenetic analysis of *L. donovani* 1S isolates compared to the global population of *L. donovani*. The three clones overlap as a single line (highlighted in red) within the 401 isolates from the Indian subcontinent (magnified section). **B** Gene coverage (copy) plotted along the 36 chromosomes in the *L. donovani* genome and colored as consistent with diploid (gray), monoploid (red), triploid (light blue) and tetraploid (dark blue). Note that Chromosomes 31 serves as a reference as it is always present as a tetraploid chromosome. Source data are provided as a Source Data file.

thickness and induration of the ear dermis (Fig. 6B, C). Notably, in contrast to mouse models, erythema was difficult to detect in the hamster ears. H&E staining of the ear dermis inoculation site however showed cellular infiltration into the tissue in the infected hamsters following injection of GLP-antigen (Fig. 6D). These results confirm that *L. donovani* Ld-Ind cultured at a one-liter scale in a bioreactor followed by production of soluble antigen under GLP compliance retained the ability to mediate an LST response in immune hamsters validating both the GLP production of soluble antigen and the hamster model.

## Discussion

As leishmaniasis elimination and control programs are advancing and new vaccines are poised to enter clinical trials, the need for the LST is essential for these initiatives to succeed. We have therefore initiated the process of re-introducing the LST through establishing and characterizing a virulent *L. donovani* strain, generating GLP-grade soluble leishmanin antigen and established mouse-CL and hamster-VL models to validate the LST response. As demonstrated within, the LST response is similar in infected and *LmCen*[−/−] vaccinated mice and hamsters. Further, the reproducible LST response observed in different laboratories located in the US and Japan in murine models of CL demonstrates that the *L. donovani* soluble antigen meets the expected manufacturing standards by regulatory bodies. Taken together, this study represents an important milestone for the re-introducing the LST in support of the control and vaccination against leishmaniasis.

Using well-established animal models of infection and immunity, it was revealed by depletion experiments (Fig. 2) that both the CD4[+] and CD8[+] T lymphocytes were necessary to generate the LST response. Likewise, in secondary infections in healed genetically resistant mice, a DTH response was shown to be mediated by both CD4[+] and CD8[+] T cells upon challenge with live promastigotes[17,18]. Immuno-histological studies in canine models following LST responses reported similar observations[24]. This is consistent with observations on the histological characterization of skin biopsies in human CL cases that showed the presence of CD4[+] and CD8[+] T lymphocytes, mononuclear phagocytes and granulocytes in both acute lesions and in the corresponding LST reactions[25]. CD8[+] T cells observed at the site of a DTH response were shown to induce TNF-α and granzyme-B that synergize with IFN-γ to activate macrophages to clear the parasites[25,26]. In the current study, biopsies of *LmCen*[−/−] immunized ear tissues showed a H&E staining indicative of robust T cell-mediated inflammatory response following injection of soluble leishmanin antigen. Immuno-histological characterization of human LST response is limited and requires further studies.

Previous studies involving the LST in areas with VL relied on leishmanin antigen derived from *L. major* or *L. infantum*[10,27,28]. The present study however chose to make leishmanin antigen from *L. donovani* to maximizing the sensitivity of the antigen for use in surveillance in VL endemic countries where humans are the only known reservoir for *L. donovani*. Further, the *L. donovani* antigen can be used

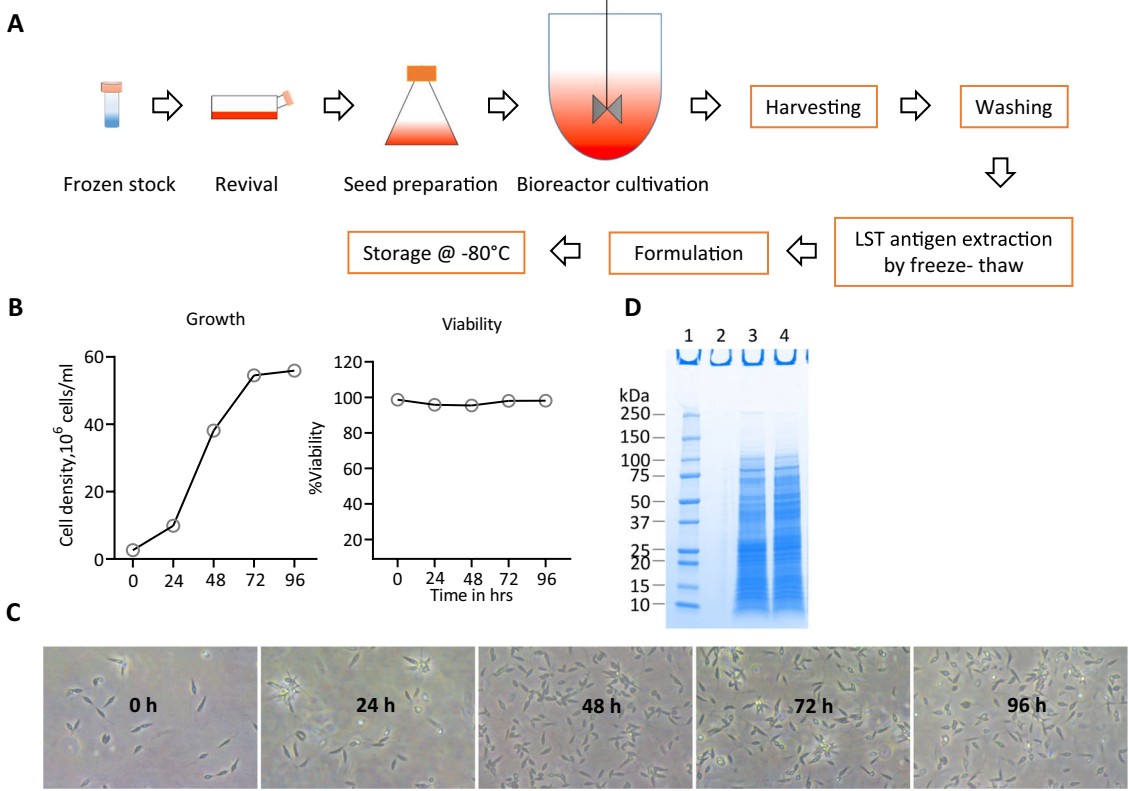

**Fig. 5 | GLP-manufacturing of *L. donovani* Ld-Ind soluble antigen. A** Workflow of LST antigen manufacturing. **B** Cell density and viability during cultivation of *L. donovani* (clone#3) promastigotes in one liter bioreactor. **C** Morphology of *L. donovani* promastigotes (400×) during bioreactor cultivation. Note: *Leishmania* promastigotes are 4–10 μm in size. **D** SDS-PAGE analysis of antigens extracted from freeze thaw cycles of *L. donovani* promastigotes grown in bioreactor. Lane 1, molecular markers, lane 3 non-reduced and lane 4, reduced form of soluble antigens. kDA, kilodalton. Source data are provided as a Source Data file.

to determine whether people vaccinated with a live attenuated *L. major*[6] in large scale human trials generate cross reactive cellular immunity against *L. donovani*. As demonstrated within, soluble antigen derived from *L. donovani* Ld-Ind generated a LST response in mice (Fig. 1) and hamsters (Fig. 6) infected with *L. major* or vaccinated with attenuated *L. major* LmCen[−/−] confirming T-cell cross reactivity with *L. donovani* antigens. Nevertheless, human studies in regions with VL should use *L. donovani* derived soluble leishmanin antigen to maximize the LST sensitivity. This study therefore focused on characterizing soluble antigen from *L. donovani* to support ongoing and future control programs and vaccine trials in areas where *L. donovani* is prevalent. It will be advantageous to integrate vaccine programs with ongoing control and elimination programs and the availability of the *L. donovani* LST will be indispensable for such combined programs going forward.

The current surveillance systems used to monitor transmission involve determining the number of VL cases in endemic and potentially endemic areas. With asymptomatic infections outnumbering VL cases by ten to one[1,29], monitoring case numbers may be ineffective in identifying new areas of transmission. Further, antibody responses lasting months are much shorter lived than the LST-associated cell-mediated immunity that can last for decades[29]. The LST is therefore a superior and necessary tool to determine ongoing and past transmission. Monitoring the age distribution of the LST response can help determine if transmission is relatively recent i.e., when a high percentage of children are positive or if there is long-standing transmission when positive cases are higher in older populations such as for example in some regions of Bhutan[30]. In instances where VL elimination is within reach, the LST will provide an effective post-elimination surveillance method in areas with new cases. In any event, it will be necessary for future surveillance and epidemiologic studies to use a

leishmanin soluble antigen made under GMP compliance derived from *L. donovani* to ensure safety and consistency between different studies. Furthermore, the soluble leishmanin antigen described within could be used in interferon-gamma release assays (IGRA) performed on blood samples that in some instances could be preferable to the LST. Studies in humans using the same *L. donovani* derived soluble antigen should compare the LST to the IGRA to establish the concordance of these assays.

The justification for using the LST as a biomarker for immunological protection against VL comes from previous surveillance studies. In Sudan, one study followed the migration of a cohort population from Western Sudan (Darfur), a region endemic for *L. major* to Eastern Sudan (Um-Salala), a region endemic for *L. donovani*[16]. Most of the migrating cohort were LST-positive due to previous CL infections caused by *L. major* in Darfur. During a three-year follow-up, only LST-negative individuals developed VL. Likewise, in a subsequent study performed in Um-Salala, Eastern Sudan, only LST-negative people developed VL during a 2-year longitudinal study comparing LST-positive and -negative cohorts[15]. In Bangladesh, like in Sudan, VL is caused by infection with *L. donovani* and again it was revealed that LST-negative but not LST-positive individuals developed VL in a longitudinal study[14]. The observations from Bangladesh support the conclusion from the Sudan studies that a positive LST is associated with protective immunity against VL supporting the argument that the LST will be essential in human vaccine trials.

Attempts have been made to use the LST in human vaccination trials[31,32]. However, the heat-killed promastigote vaccine used in these studies had poor efficacy and it was therefore difficult to establish whether the LST correlated with immunological protection following vaccination. The animal models used in this study did however reveal that immunological protection following vaccination was associated

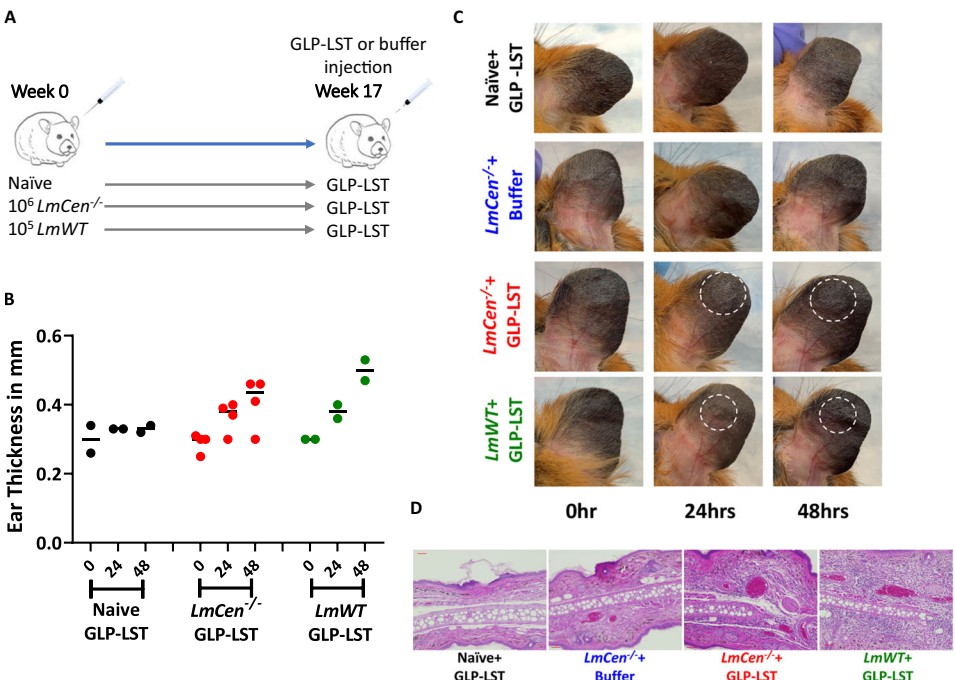

**Fig. 6 | LST as a surrogate of immunogenicity of attenuated *LmCen⁻/⁻* and wild-type *L. major* in hamsters. A** Schematic diagram showing the schedule of immunization with attenuated *LmCen⁻/⁻* or wild-type *L. major* followed by inoculation with GLP-soluble LST antigens in hamsters. GLP, Good laboratory practices. LST, Leishmanin skin test. **B** Ear thickness following injection of GLP-soluble LST antigen in the contralateral ear of Naïve, attenuated *LmCen⁻/⁻* or wild-*type L*. major infected hamsters at 0, 24 and 48 h after injection. Results are shown as mean values, (*n* = 2 for naïve-GLP-LST; *n* = 3 for *LmCen⁻/⁻* GLP-LST; *n* = 2 *LmWT* GLP-LST groups. Data are the representation of two biologically independent experiments. **C** Representative images of the contralateral ears of hamsters injected with GLP-soluble LST antigens or excipient solution (buffer) 48 h post-injection. **D** H&E staining of the ear sections of naïve, attenuated *LmCen⁻/⁻* immunized or wild-*type L*. major infected hamsters following GLP-soluble LST antigen or excipient inoculation showing cellular infiltration. Scale bar 50 µm. Source data are provided as a Source Data file.

with the LST response justifying its use in future human vaccine trials. Taken together, the present study addresses the major bottleneck in re-introducing the LST through the generation of GLP-grade soluble antigen from a well-defined virulent laboratory strain of *L. donovani* and validating the LST in animal models. We are currently producing GMP-grade LST antigen for toxicology studies followed by validation studies in individuals from endemic countries who have been treated and cured of VL who now have protective cellular immunity against *L. donovani*.

As there is currently no GMP-grade LST antigen being produced, we will make the LST antigen described within widely available for use in endemic countries, following validation studies and regulatory approvals.

## Methods
### Study design and ethical statement
All research complies with all relevant ethical regulations. Animal experiments in this study were reviewed and approved by the Animal Care and Use Committee of the Center for Biologics Evaluation and Research, U.S. Food and Drug Administration (ASP-1999#23 and ASP-1995#26) and the National Institute of Allergy and Infectious Diseases (NIAID/NIH) (http://grants.nih.gov/grants/olaw/references/phspolicylabanimals.pdf) under animal protocol LMVR4E, and Nagasaki University for ethics on animal experiments (approval number 2004271624, 2004271625, 2104011711) and recombinant DNA experiments (1902201550, 2103311722). The NIAID DIR Animal Care and Use Program complies with the Guide for the Care and Use of Laboratory Animals and with the NIH Office of Animal Care and Use and Animal Research Advisory Committee guidelines. The housing conditions of animals were followed standard guidelines by NIH guidelines for the humane care and use of animals. Immunization infections with

*LmCen⁻/⁻* parasites were performed in a hamster model of VL or a mouse model of CL to determine the potency of the LST response against soluble antigen made from *L. donovani* Ld-Ind. No statistical method was used to predetermine sample size. No data were excluded from the analyses. The experiments were not randomized and the Investigators were not blinded to allocation during experiments and outcome assessment.

### Parasites
*L. donovani* Ld-Ind is an isolate originally derived from the human bone marrow of a visceral leishmaniasis patient from West Bengal in 1993 and given the WHO designation code; MHOM/IN/93/BI2301/LRC-751. The *L. donovani* and *L. major centrin* gene-deleted *LmCen⁻/⁻* promastigotes were cultured as previously described[6].

### Animals
Six to eight-week-old female outbred Syrian golden hamsters (*Mesocricetus auratus*) were obtained from the Harlan Laboratories Indianapolis, USA. All animals were housed either at the Food and Drug Administration (FDA) animal facility, Silver Spring (MD) or the National Institute of Allergy and Infectious Diseases (NIAID), Twin-brook campus animal facility, Rockville (MD), under pathogen-free conditions.

Female 6- to 8-wk-old C57BL/6 mice were immunized and/or leishmanized with $1 \times 10^6$ total stationary phase *LmCen⁻/⁻* or 1000 *L. major* wild-type (FV9) parasites by intradermal injection in the left ear in 10 µl PBS. After 12 weeks post-infection, healed mice were inoculated on the contralateral ear with GLP-LST by needle inoculation.

### Sand Fly Infections
Colony-bred 2- to 4-day-old *Lutzomyia longipalpis* females were infected by artificial feeding on defibrinated rabbit blood (Spring

Valley Laboratories, Sykesville, MD) containing $5 \times 10^6$ amastigotes or first-passage promastigotes and 30 μL penicillin/streptomycin (10 000 units penicillin/10 mg streptomycin) per mL of blood for 3 h in the dark. Fully blood-fed flies were separated and maintained at 26 °C with 75% humidity and were provided 30% sucrose.

## Pre-transmission scoring

*L. donovani*-infected sand flies were scored at days 8 and 11, respectively, to assess pre-transmission infection status. Flies (7–13) were washed, and each midgut was macerated with a pestle (Kimble Chase, Vineland, NJ) in an Eppendorf tube containing 50 μL of PBS. Parasite loads and percentage of metacyclics per midgut were determined using hemocytometer counts. Metacyclics were distinguished by morphology and motility.

## Transmission of *Leishmania* to Hamsters via Sand Fly Bites

Hamsters were anesthetized intraperitoneally with ketamine (100 mg/kg) and xylazine (10 mg/kg). Ointment was applied to the eyes to prevent corneal dryness. Flies (20–30) with mature infections were applied to both ears of each hamster through a meshed surface of vials held in place by custom-made clamps. The flies were allowed to feed for 1–2 h in the dark. Hamsters were monitored daily for appearance, activity, swelling, pain, and ulceration during the course of infection, and their body weights were recorded weekly. The end point for this study was reached when hamsters exhibited any of the following criteria: a 20% weight loss; inability to eat or drink; or becoming non-responsive to external stimuli.

## Post-transmission scoring

The number of blood-fed flies was determined post-transmission. Where the infection status was established, all blood-fed flies were dissected and examined. Flies were assigned a "transmissible infection" status when a mature infection contained numerous active metacyclic promastigotes.

## LST by intradermal inoculation

Contralateral ears of immunized mice were injected with Leishmanin antigens intradermally using 28 or 29 G needle. The human LST typically used 50 μg[10], therefore we chose a 10 fold lower amount (5 μg in 10 μl) in the animal models. Since this amount provided consistent responses, all studies used 5 μg. Ear thickness of the injected mice was measured at 24 and 48 h post-injection. Results are expressed as mean ± SEM and data reported from two independent laboratories (Nagasaki and FDA). Statistical analysis was performed by unpaired two-tailed t-test.

## Immunohistochemistry

At 48 h post-injection, animals were euthanized and respective ears were collected. Ears were stored in 10% Formalin-phosphate buffered solution for at least 48h, followed by 70% Ethanol till further processing for paraffin embedding and sections. Some of the sections were stained with Hematoxylin and Eosin (H&E) and other sections were processed for immunostaining. Recombinant Anti-CD4 (ab183685) and anti-CD8 (ab209775) antibodies were used to stain CD4$^+$ or CD8$^+$ T cell staining. All histochemical staining was performed at Histoserv, USA and the Biomedical Research Support Center (BRSC), Nagasaki University School of medicine, Japan.

## CD4$^+$/CD8$^+$ T cell depletion

Depletion of CD4$^+$ or CD8$^+$ T cells was accomplished with a single intraperitoneal injection of either anti-mouse CD4 mAb (clone GK1.5, 500 μg/100 μL, #BE0003-1) or anti-CD8 mAb (clone YTS169.4, 300 μg/100 μL, #BE0117), whereas the non-depleted control group was injected with saline. Leishmanin antigens were injected 36 h post-depletion.

## Parasite DNA extraction and whole genome sequencing

DNA from promastigote cultures was extracted using a Dneasy column according to the manufacturer's instruction (Qiagen). PCR-free library preparation (Lucigen) and 6000 sequencing (Illumina) was performed at CBER-FDA biotechnology core. Raw reads were processed as described[21]. Briefly, Illumina paired reads were aligned to the reference *L. donovani* LdCL reference genome sequence obtained from TriTrypDB[33] using the Burrows-Wheeler Aligner (version 0.7.17[34]), file formats transformed using samtools (version 1.10[35]), and variant calling was done with VarScan2 (version 2.4.3[36]) to generate VCF files. Per sample candidate SNP were called by VarScan2 with a minimum coverage of 0.4× mean genome coverage, a minimum alternate allele frequency of 20% (read/read), a minimum average base quality of 15 across the reads and a 90% significance threshold. For phylogeny generation, additional sequences previously analyzed[21] and obtained from GenBank from whole-genome sequencing projects of *L. donovani* were included along with *L. donovani* Ld-Ind clones to populate the phylogeny and identify the origins of the Ld-Ind clones. Gene coverage was plotted using Circos (version 0.69-8[37]) to determine copy and chromosome number changes. Intersections of variant calls of clones 1,3 and 4 were performed using bcftools (version 1.16[38]) and variants private only to one or two of the Ld-Ind clones were analyzed and reported if found.

## Production of leishmanin antigen

A frozen vial of *L. donovani* Ld-Ind (Clone #3) promastigotes was thawed, and the seed was prepared in M199 based growth medium in a T-flask and shake flask. The seed was inoculated into a 2 L bioreactor containing 1 L of growth medium. The promastigotes were cultivated in the bioreactor for 96 h. Promastigote growth and morphological characteristics were monitored (Fig. 6B, C). Promastigote parasites were harvested after 4 days of bioreactor cultivation when cultures reached the stationary phase of growth. *L. donovani* promastigotes were harvested and washed six times by centrifugation at 1300*g* for 5 min using sterile 1× phosphate buffered saline (PBS) with 1% (w/v) glucose. LST soluble antigen was extracted by freeze-thaw method as per published protocol where final washed promastigotes pellet was re-suspended into 5 volumes of sterile water (Reed et al.[39]). This solution was freeze-thawed 12 times by freezing it in liquid nitrogen and thawing in 37 °C water bath. The disrupted promastigotes were diluted in 10 volumes of 1× PBS and centrifuged at 10,000*g* for 30 min at 4 °C. The total protein concentration of the soluble Leishmania extract (Leishmanin soluble antigen) was determined by BCA protein assay. The estimated LST antigen concentration was 1.8 mg/ml. SDS-PAGE was run for extracted LST antigen and a standard band pattern of LST antigen was observed as given in Fig. 6D. The LST antigen was adjusted to 0.5 mg/ml with sterile PBS containing Tween 80 and phenol, at final concentration of 0.0005% (w/v) and 0.28% (w/v) respectively. Further, the formulated LST antigen was sterilized with 0.2 μm filter and stored at −80 °C.

## Ethical statement

The animal protocol for this study has been approved by the Institutional Animal Care and Use Committee at the Center for Biologics Evaluation and Research, US FDA (ASP 1995#26). The animal protocol is in full accordance with "The guide for the care and use of animals as described in the US Public Health Service policy on Humane Care and Use of Laboratory Animals 2015". Animal experimental procedures performed at the National Institute of Allergy and Infectious Diseases (NIAID) were reviewed by the NIAID Animal Care and Use Committee under animal protocol LMVR4E. The NIAID DIR Animal Care and Use Program complies with the Guide for the Care and Use of Laboratory Animals and with the NIH Office of Animal Care and Use and Animal Research Advisory Committee guidelines. Detailed NIH Animal Research Guidelines can be accessed at https://oma1.od.nih.gov/

manualchapters/intramural/3040-2/. Animal experimental procedures performed at Nagasaki University were approved by the Institutional Animal Research Committee of Nagasaki University (No.1606211317 and 1505181227), the Nagasaki University Recombinant DNA Experiments Safety Committee (No. 1403041262 and 1407221278), and performed according to Japanese law for the Humane Treatment and Management of Animals.

## Statistical analysis
Statistical analysis of differences between means of groups was determined Student's *t* test using Graph Pad Prism 7.0 software. The statistical tests and the significance values are described in the figure legends.

## Reporting summary
Further information on research design is available in the Nature Portfolio Reporting Summary linked to this article.

## Data availability
The sequencing data for the Ld-Ind-clone1,3 and 4 have been deposited at Genbank's sequencing read archive (SRA) as BioSamples SAMN31408468, SAMN31408469 and SAMN31408470 respectively, under the PRJNA893015 BioProject. The LdCL reference genome is available at TriTrypDB [https://tritrypdb.org/tritrypdb/app/record/dataset/TMPTX_IdonCL-SL]. Flow cytometry data has been deposited at FlowRepository under accession FR-FCM-Z6UG. Source data are provided with this paper.

## Code availability
All used computer code has been previously published and is detailed in Methods.

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

## Acknowledgements
Funding was provided from the Global Health Innovative Technology Fund, G2019-213R1 (to A.R.S.), the Canadian Institutes of Health Research, 153282 (to G.M.) and 187858 (to P.L.), intramural funding from CBER, FDA (to H.L.N.), the Intramural Research Program of the NIH, National Institute of Allergy and infectious diseases (S.K., J.V.) and the Fonds de recherche du Quebec-Sante (to K.P.). The findings of this study are an informal communication and represent the authors' own best judgments. These comments do not bind or obligate the Food and Drug Administration.

## Author contributions
R.D., J.A., L.K., P.L., Subir Karmakar, and K.P.S. performed the experiments and analyzed the data and edited the manuscript. P.L., Swarnendu Kaviraj, and K.P. analyzed the data and edited the manuscript. Shaden Kamhawi, J.V., and S.S. edited the manuscript. S.H., A.R.S., S.G., H.L.N., and G.M. were responsible for funding acquisition, supervising, writing, and editing the manuscript.

## Competing interests
The authors declare the following competing interests: The FDA is currently a co-owner of two US patents that claim attenuated Leishmania species with the Centrin gene deletion (US7,887,812 and US 8,877,213). K.P.S., Subir Karmakar, Swarnendu Kaviraj and S.S. are employees of Gennova Biopharmaceuticals. The remaining authors declare no competing interests.
