## [Peer Review File · Nature Communications]

REVIEWER COMMENTS

Reviewer #1 (Remarks to the Author):

Ranadhir Dey et al. submitted the manuscript entitled "Production of a Leishmanin Skin Test antigen from *Leishmania donovani* and validation in animal models: Progress toward reintroduction in the field". The authors explained about the production of a new leishmanin for LST prepared of *L. donovani* Ld1S. The study was designed and completed very precisely and almost covered every aspect of leishmanin preparation. Any effort to improve leishmaniasis management should be fully encouraged, I have a few comments as follow:

Major comments:

Previously, several leishmanin types were produced and used in new and old world, the last one was produced from *Leishmania major* under support and approval of TDR/WHO, *Leishmania major* which is the same isolate used for leishmanization of more than 2 million people and etc. the authors should explain and justify why *Leishmania donovani* strain was used to develop leishmanin, although strong cross reaction exists. The authors did not mention any references from old leishmanins, and why need new ones. It is need to have a comparison with the leishmanin produced from *Leishmania major* and used in humans.

Minor comments:

Line 28, the statement of "Due to a lack of antigen used for the intradermal injection, the LST is no longer available" seems not to be correct, Pasteur Institute of Iran produces leishmanin, I have contacted Prof. Alimohammadian who is responsible for leishmanin production at Pasteur Institute and he said leishmanin is available but there is no demand.

Line 57-63 needs reference

Line 64, the statement "VL is potentially a vaccine-preventable disease" Instead of VL, it is more scientific to use leishmaniasis since there are more evidence that CL is preventable by vaccine.

Line 84-86, the statement about leishmanin reference of Skraba et al. is about Montenegro skin test, the authors should explain about old world leishmanin and use the references in this regard, there are plenty of references of using leishmanin in vaccine clinical trials, unfortunately none is included.

Line 88-89, the statement of "epidemiologic studies have shown that a positive LST is associated with long-lasting protective immunity" seems not to be correct, LST positivity is an indication of previous exposure to *Leishmania* antigens and not an indication of protection, in two vaccine trial studies conducted and supported by TDR/WHO, participants were leishmanin skin tested prior to vaccination and LST positive individuals were excluded, the follow up of LST positive individuals (not vaccinated) showed that the incidence rate of CL in the leishmanin positive individuals was not significantly different from LST negatives ones, although in ZCL endemic area the rate of CL was more in LST positives.

Line 92, the statement of “The LST is better than serology... “ I need to say that serology in old world CL is not practical since the antibody titer is usually not detectable, so serology could not be used to identify exposure to Leishmania.

5µg of Leishmanin antigens was used but the authors did not explain why only 5µg was tested, is there any data on dose response to show that 5ug is superior.

Reviewer #2 (Remarks to the Author):

This is extensive and excellent article on the preparation and testing of a crude DTH skin test in the mouse model, aiming the evaluation of *Leishmania donovani* infection or vaccines. The authors prepared *L. donovani* soluble antigen for *L. major* wild-type and centrin -/- infection challenge and evaluated the requirement of CD4+ and CD8+ cells.

L. donovani antigen originated from an established isolate from hamsters infected by sandflies.

The authors demonstrated the feasibility and stability of their antigen and demonstrated the essential roles of CD4+ and CD8+ cells, as expected. The study is a clear demonstration of GMP for DTH antigen production.

However, the use of *L. donovani* antigen for infection caused by *L. major* and the use of a species that is not a natural reservoir limit the application for humans and reservoirs. Moreover, if the paper is to establish a standard model for GMP for leishmanin production, due to its complexity, it might hamper the production of other types of antigen. This is due to the possibility of much simpler methods being developed, without the use of transgenic *Leishmania* species as a challenge. *L. donovani* and *L. infantum* would be the preferred infection if aimed at humans and dogs.

Therefore, I suggest that the authors address the issue if they are trying to establish a standard GMP leishmanin DTH production and testing for humans and animal reservoirs.

Reviewer #3 (Remarks to the Author):

The study entitled “Production of a Leishmanin Skin Test antigen from *Leishmania donovani* and validation in animal models: Progress toward reintroduction in the field” aims at describing the production of a *Leishmania* parasite lysate under GLP condition, that may be used for the study of the immune response in surveillance or vaccine studies. While this work is technically sound and very well performed, it derives from decades of using such *Leishmania* skin test (LST) antigen (Carstens-Kass et al, 2021; Pacheco-Fernandez et al, 2021) and brings no new knowledge to the field. Indeed, the usefulness of a standardized LST antigen for research has been known for a long time, as well as its limitations

(Boroujeni et al, 2013). Importantly, there have been previous attempts at providing a standardized LST, that have failed for mostly non-scientific reasons (Carstens-Kass et al, 2021), and how these issues will be solved for this new preparation is not addressed in this study.

No information is provided on the origin of the *L. donovani* strain Ld1S selected for the LST antigen (Source of the isolate (patient or else), year, etc...). No information is provided on the reproducibility of the antigen preparation from different batches of GLP production, nor on the composition of the mixture, which may be some of the novel aspects of the work. Some proteomic data or similar would be key as well as potential issues with the stability/reproducibility of the preparation. And most importantly, how this new preparation will be distributed or made available to the Leishmania research community is not addressed.

Pacheco-Fernandez T, G Volpedo, S Gannavaram, P Bhattacharya, R Dey, A Satoskar, G Matlashewski, HL Nakhasi (2021) Revival of Leishmanization and Leishmanin. *Front Cell Infect Microbiol* doi: 10.3389/fcimb.2021.639801

Carstens-Kass J, K Paulini, P Lypaczewski, G Matlashewski (2021) A review of the leishmanin skin test: A neglected test for a neglected disease. *PLoS Negl Trop Dis* 15(7):e0009531.

doi: 10.1371/journal.pntd.0009531.

Boroujeni AM, M Aminjavaheri, B Moshtagian, A Momeni, AZ Momeni (2013) Reevaluating leishmanin skin test as a marker for immunity against cutaneous leishmaniasis. *Int J Dermatol* 52(7):827-30. doi: 10.1111/j.1365-4632.2012.05850.x

NCOMMS-23-03512: Point by point responses to reviewers comments

Production of a Leishmanin Skin Test antigen from *Leishmania donovani* and validation in animal models: Progress toward reintroduction in the field.

Reviewer 1. Major Comment

The authors should explain and justify why *Leishmania donovani* strain was used to develop leishmanin, although strong cross reaction exists. The authors did not mention any references from old leishmanins, and why need new ones. It is need to have a comparison with the leishmanin produced from *Leishmania major* and used in humans.

Response: We appreciate this comment. We have now provided justification in the **Discussion**, page 10, second paragraph that explains why the LST leishmanin antigen was produced from *L. donovani*. We have also added original references for the *L. major* derived leishmanin antigen produced in Iran (Alimohammadian 1993) in this revised paragraph.

Reviewer 1. Minor Comment

Line 28, the statement of “Due to a lack of antigen used for the intradermal injection, the LST is no longer available” seems not to be correct, Pasteur Institute of Iran produces leishmanin, I have contacted Prof. Alimohammadian who is responsible for leishmanin production at Pasteur Institute and he said leishmanin is available but there is no demand.

Response: We likewise contacted Professor Alimohammadian before we started this project and were informed that antigen from *L. major* production stopped in 2018 and that the stock of vials remaining expired in Feb 202. In any event, GMP grade leishmania antigen is no longer being produced under GMP conditions in Iran or anywhere else in the world. We have also been in discussion with the WHO NTD department and other stake holders in Asia and Africa who all agree there is an urgent unmet need for GMP grade LST leishmanin antigen.

Line 57-63 needs reference:

Response: We have added 2 references for this information in the **Introduction**, second paragraph).

Line 64, the statement “VL is potentially a vaccine-preventable disease” Instead of VL, it is more scientific to use leishmaniasis since there are more evidence that CL is preventable by vaccine.

Response: We have made this change in the **Introduction**, first sentence of the third paragraph that now states “Leishmaniasis”.

Line 84-86, the statement about leishmanin reference of Skraba et al. is about Montenegro skin test, the authors should explain about old world leishmanin and use the references in this regard, there are plenty of references of using leishmanin in vaccine clinical trials, unfortunately none is included.

Response: We have included references (Khalil 2000, Mayrink 1979) for the use of the LST in old and new world leishmaniasis vaccine trials as requested in the **Discussion**, last paragraph (page

12). We believe it is more appropriate to include this information in the **Discussion** rather than the **Introduction** that refers to the more general use of the DTH to determine pre-exposure.

Line 88-89, the statement of “epidemiologic studies have shown that a positive LST is associated with long-lasting protective immunity” seems not to be correct, LST positivity is an indication of previous exposure to Leishmania antigens and not an indication of protection, in two vaccine trial studies conducted and supported by TDR/WHO, participants were leishmanin skin tested prior to vaccination and LST positive individuals were excluded, the follow up of LST positive individuals (not vaccinated) showed that the incidence rate of CL in the leishmanin positive individuals was not significantly different from LST negatives ones, although in ZCL endemic area the rate of CL was more in LST positives.

Response: We agree that the evidence for LST associated with protection against CL not strong. However, it is strong for VL. This why we stated on line 90 “immunity against reinfection with *Leishmania*, most notably VL”. The 3 epidemiological papers referenced (Bern 2006, Khalil 2002, Zilstra 1994) all clearly show LST associated with protection against VL. We have now further clarified this issue by stating “long lasting protective immunity against VL” (**Introduction**, page 4, first paragraph) which is correct and consistent with previous studies.

5µg of Leishmanin antigens was used but the authors did not explain why only 5µg was tested, is there any data on dose response to show that 5ug is superior.

Response: The human LST typically used 50 ug, therefore we chose a 10 fold lower amount (5 ug) in the animal models. Since this amount provided consistent responses, all studies used 5 ug. This is now stated in the **Methods**, page 13.

Reviewer 2

The use of *L. donovani* antigen for infection caused by *L. major* and the use of a species that is not a natural reservoir limit the application for humans and reservoirs.

Response: This comment is somewhat ambiguous. Humans are indeed the only known reservoir for *L. donovani* and Leishmanin antigen prepared from *L. donovani* did generate an LST response in *L. donovani* infected hamsters (Fig.6) and *L. major* infected mice (Fig. 1) confirming cross reactivity in VL and CL models. As indicated throughout the paper, the priority will be to use the *L. donovani* derived leishmanin LST antigen for VL surveillance and VL vaccination studies for protection against *L. donovani*. This issue has now been clarified in the **Discussion** on page 10, second paragraph.

If the paper is to establish a standard model for GMP for leishmanin production, due to its complexity, it might hamper the production of other types of antigen. This is due to the possibility of much simpler methods being developed, without the use of transgenic *Leishmania* species as a challenge.

Response: Any leishmanin antigen used in future humans studies must meet GMP standards as argued in this paper and this is now required by regulatory authorities in all endemic countries. We agree it may be necessary to make antigens from different species such as for Old and New

World species. As argued within, the priority is to develop an antigen from *L. donovani* to focus on supporting VL elimination programs in Asia and Africa and vaccine trials against VL.

I suggest that the authors address the issue if they are trying to establish a standard GMP leishmanin DTH production and testing for humans and animal reservoirs.

Response: The priority is to develop a GMP antigen for the LST in humans as described. However, the reviewer raises the possibility of also using the leishmanin antigen for testing in animal reservoirs. Although we have considered this possibility, having a positive LST would not necessarily provide evidence of as reservoir. We therefore would rather not include this possibility in the revised paper.

Reviewer 3

There have been previous attempts at providing a standardized LST, that have failed for mostly non-scientific reasons (Carstens-Kass et al, 2021), and how these issues will be solved for this new preparation is not addressed in this study.

Response: We agree that previous attempts to provide a standardized LST antigen have been largely non-scientific. Interestingly, the Reviewer refers to our recent review papers (Carstens-Kass 2021, Pacheco-Fernandez 2021) to illustrate this point. These papers also argue that there is a need for standardized LST antigen from *L. donovani*. This study describes the first LST antigen produced in a biopharmaceutical facility (Genova, Pune India) that has FDA and WHO certification for the production of GMP products. We therefore believe that this study does begin to address the non-scientific reason that the LST antigen is not available.

No information is provided on the origin of the *L. donovani* strain Ld1S selected for the LST antigen (Source of the isolate (patient or else), year, etc...).

Response: As the *L. donovani* strain Ld1S has been isolated prior to the initiation of this study, it was necessary to rederive it from an infected sandfly and confirm its virulence in visceral infection in hamsters as shown in Figure 3. This represents a long term and expensive process. Further the geographic origin was precisely determined by whole genome sequence and phylogenetic analysis as shown in Figure 4. These analysis provide a highly detailed characterization of the *L. donovani* strain acceptable for GLP and GMP production of antigen.

No information is provided on the reproducibility of the antigen preparation from different batches of GLP production, nor on the composition of the mixture, which may be some of the novel aspects of the work. Some proteomic data or similar would be key as well as potential issues with the stability/reproducibility of the preparation. And most importantly, how this new preparation will be distributed or made available to the Leishmania research community is not addressed.

Response: We are aware of these issues identified by the reviewer. However, these issues are largely addressed and reviewed through regulatory agencies and are beyond the scope of the present study. We have also added a sentence at the end of the **Discussion** confirming that the LST antigen will be widely available in endemic countries once it has been approved by regulatory agencies.

REVIEWER COMMENTS

Reviewer #2 (Remarks to the Author):

In the manuscript still remains the bias suggests that *L. donovani* antigen would be better than *L. major*, particularly regarding zoonotic visceral leishmaniasis where the agent is *L. infantum*.

Also, remains a long jump from testing a vaccine with *L. major* for a disease whose agent is a different species.

By the way, the authors should refer to Leishmaniasis (several diseases) rather than Leishmaniasis suggesting it is a single nosological entity.

Reviewer #3 (Remarks to the Author):

Unfortunately, it seems that the authors failed to address the issues raised previously. As mentioned before, the production standardized LST antigen is not novel, nor its immunological properties or intended use, and the author's claim that "This study describes the first LST antigen produced in a biopharmaceutical facility" is an overstatement as many of the key aspects of such a process are not described in the paper as they consider these beyond the scope of the study. As mentioned before, no information is provided on the reproducibility of the antigen preparation from different batches of GLP production, nor on the exact composition of the mixture, which may be some of the novel aspects of the work and would allow to assess the stability/reproducibility of the preparation. These are indeed key and relevant aspects pertaining to the GLP/GMP manufacture and at least some of these data, which the authors mention are under review by regulatory agencies, should be presented.

Also, the addition of a sentence stating that the "LST antigen will be widely available" is only wishful thinking as the same was said of previous standardized LST preparations which are no longer available and none of the key issues preventing availability of these preparations is discussed, contrary to what the authors state.

Regarding the *L. donovani* Ld1S strain, I understand that they ensured infectivity and virulence by hamsters/sandflies infections, even though the strain may have been isolated much prior to this study. However, any general information on the history of this strain would still enrich the paper (year isolated, from a patient or an animal host, etc..?)

Reviewer comments

Reviewer #2 (Remarks to the Author):

In the manuscript still remains the bias suggests that *L. donovani* antigen would be better than *L. major*, particularly regarding zoonotic visceral leishmaniasis where the agent is *L. infantum*.

Response: We are unable to understand the reviewer's intent. For example, we think he/she wants a justification for using antigen from *L. donovani* instead of *L. major* as a biomarker of efficacy in future vaccine studies in humans. Our goal is to target the *LmCen*^{-/-} vaccine first against visceral leishmaniasis caused by *L. donovani*/*L. infantum* because it is potentially fatal disease. The rationale for using an attenuated *L. major LmCen*^{-/-} based vaccine for VL is justified by published experimental and epidemiological studies that show a prior *L. major* infection confers durable cross protection against VL. Secondly, LST is a reliable surrogate of protection against VL but its value as a predictor of immunity against CL is not clear as indicated by the literature in this regard. Further during our ongoing consultation with WHO-NTD regarding the production of GMP grade Leishmanin for LST, the experts affirmed the suitability of *L. donovani* as the source for LST material for surveillance in VL endemic countries. For the above mentioned reasons, it remains appropriate and justifiable to use LST antigen made from a related *L. donovani* species rather than *L. major*. Indeed we did state in the Discussion (Page 10, second paragraph, lines 259-261) that the priority is to use the LST for surveillance in countries with VL caused by *L. donovani* and that the LST will be used as surrogate marker for vaccination efficacy against VL cause by *L. donovani*.

Also, remains a long jump from testing a vaccine with *L. major* for a disease whose agent is a different species.

Response: We differ here in our opinion from the reviewer: "it is a long jump to test a vaccine derived from *L. major* for protection against a different species". The literature does not support this comment by the reviewer based on our published studies (ref 23) and epidemiological studies by others (ref 14, 15, 16) as detailed in lines 288-299.

By the way, the authors should refer to Leishmaniasis (several diseases) rather the Leishmaniasis suggesting it is a single nosological entity.

Response: Page 3, first word of the second paragraph, the word "Leishmaniasis" has been changed to "Leishmaniasis". Throughout the paper, we also distinguish that visceral leishmaniasis and cutaneous leishmaniasis are different diseases cause by different species of *Leishmania* parasites.

Reviewer #3 (Remarks to the Author):

Unfortunately, it seems that the authors failed to address the issues raised previously. As mentioned before, the production standardized LST antigen is not novel, nor its immunological properties or intended use, and the author's claim that "This study describes the first LST antigen produced in a biopharmaceutical facility" is an overstatement as many of the key aspects of such a process are not described in the paper as they consider these beyond the scope of the study. As mentioned before, no information is provided on the reproducibility of the antigen preparation from different batches of GLP production, nor on the exact composition of the mixture, which may be some of the novel aspects of the work and would allow to assess the stability/reproducibility of the preparation. These are indeed key and relevant aspects pertaining to the GLP/GMP manufacture and at least some of these data, which the authors mention are under review by regulatory agencies, should be presented.

Response: GLP-LST antigens were produced from four production lots as per the guidelines from the regulatory guidance documents. As requested, data related to reproducibility of the parasite growth characteristics, and antigen preparation from three additional lots are included as supplementary information (new supplementary Figure 5) and on page 8, second paragraph.

Also, the addition of a sentence stating that the "LST antigen will be widely available" is only wishful thinking as the same was said of previous standardized LST preparations with are no longer available and none of the key issues preventing availability of these preparations is discussed, contrary to what the authors state.

Response: This is a relevant issue, however, we have previously covered in depth in a review paper detailing the key issues preventing availability (ref 10) and it is therefore not necessary to present these details again in this publication. This is emphasized in the introduction (page 4, second paragraph, last sentence, lines 92-95 including ref 10.) We do not agree with the comment about "wishful thinking", otherwise the efforts described in this paper would not have been undertaken

Regarding the *L. donovani* Ld1S strain, I understand that they ensured infectivity and virulence by hamsters/sandflies infections, even though the strain may have been isolated much prior to this study. However, any general information on the history of this strain would still enrich the paper (year isolated, from a patient or an animal host, etc..?)

Response: As requested by this reviewer, we are providing additional information about the *L. donovani* isolate used to derive the Ld1S clones used to make the leishmania antigen. This new information is presented in Methods, page 13.

REVIEWERS' COMMENTS

Reviewer #3 (Remarks to the Author):

In spite of what the authors claim, the issue of availability of the preparation is not addressed in the manuscript, nor in previous publications. Thus, I would recommend deleting the last sentence: "GMP grade leishmanin antigen from *L. donovani* will be made widely available for use in endemic countries once it has been approved by necessary regulatory agencies" as there is no clear plan for this to occur.